# Axial Load Measurement of Bolts with Different Clamping Lengths Based on High-Frequency Ultrasonic ZnO Film Sensor

**DOI:** 10.3390/s23073712

**Published:** 2023-04-03

**Authors:** Xuechao Jing, Hongwei Dai, Wanli Xu, Yue Zhao, Jun Zhang, Bing Yang

**Affiliations:** 1School of Power and Mechanical Engineering, Wuhan University, Wuhan 430072, China; jxchao2018@gmail.com (X.J.); wanlixu_ucas@163.com (W.X.);; 2International Joint Research Center for Surface and Interface Materials Science and Engineering, Wuhan University, Wuhan 430072, China

**Keywords:** ultrasound, acoustoelasticity, ZnO piezoelectric film, bolt load measurement, effective clamping length

## Abstract

The ultrasonic testing method has been widely used for measuring the axial load of bolts. However, systematic calibrations are prerequisite if specific bolts have different clamping length configurations, which leads to low efficiency and measurement errors. The focus of this work was to measure the axial load of bolts with different clamping lengths by proposing a method of clamping length correction based on piezoelectric films in order to avoid the complicated calibration steps. Firstly, the relationship between longitudinal wave time-of-flight (TOF) and axial load under different clamping lengths was studied to correct the difference between the effective stress length and the actual clamping length. Secondly, the high-frequency ZnO piezoelectric film sensor was fabricated on the bolts to improve the accuracy of longitudinal wave TOF measurement. The results showed that the center frequency of the fabricated ultrasonic sensor reached 25 MHz, which could realize the high precision measurement of TOF. The proposed correction model proved to be effective for decreasing the measurement error below 2.7% in this experiment. In conclusion, the proposed method simplified the calibration procedure for different application configurations of the same bolt and realized the efficient measurement of bolt axial load.

## 1. Introduction

The bolted connection structure has the functions of fastening, connecting, and sealing. It plays an indispensable role in various mechanical structures and has become the most widely used mechanical part. The reliability of bolted connections is the premise of the safe operation of industrial components. The vibration, external impact, and temperature change of a bolt in service will lead to changes in preload, affect its safety performance, and eventually lead to bolt relaxation and failure. Therefore, it is necessary to control the preload of bolts and monitor the axial load variation of service bolts.

At present, the commonly used axial load measurement methods include the torque method, turn-of-the-nut method, FBG (Fiber Bragg Grating) method, and ultrasonic method [1,2]. The torque and turn-of-the-nut methods are always used for pre-tightening, and their axial load measurement error is up to 25% [3]. The FBG method generally encapsulates the FBG sensor on the outer surface of the screw or inserts it inside the bolt to measure the unevenly distributed strain [4]. However, the bolt structure needs to be destroyed to measure the axial load of the bolt [5]. Ultrasonic testing is considered to be the load measurement method with the most potential due to its high accuracy and non-destruction of bolt structure [6,7].

The ultrasonic load measurement method is based on the acoustoelastic effect, which means that the velocity of the ultrasonic wave can vary according to the stress state [8,9]. Single-mode wave and bi-mode wave are the two main methods [10,11]. The single-mode wave method requires a calibration to establish the linear relationship between the bolt load and time-of-flight (TOF) [12,13,14]. However, calibrations for specific configurations are prerequisite to obtain model parameters under stress-free conditions. The influence of stress on the ultrasonic velocity is dominant in the clamping zone of the bolt. Different clamping lengths will lead to changes in the TOF of the same bolt with the same load. Systematic calibration is needed for high accuracy measurement, which leads to low efficiency for industrial applications of bolts with different tightening configurations.

In order to reduce the number of calibration steps, the bi-mode wave method combines longitudinal and transverse waves for the measurement of axial load. Chaki pointed out that the ratio of the TOF of longitudinal and transverse waves is linear to the axial load of the bolt. By introducing an effective stress length factor, he proposed a simplified calibration method using only two clamping length configurations [15]. However, this simplified calibration method has a high requirement for the accurate measurement of both longitudinal and transverse wave TOFs. When using an ordinary piezoelectric transducer, it takes multiple echoes to calculate the flight time in order to reduce the error induced by the couplant. Some other ultrasonic transducers can obtain both longitudinal wave and shear wave TOFs, such as the electromagnetic acoustic transducer (EMAT) [16,17]. In addition, the linear relationship between the TOF ratio of the mode-converted shear wave and the axial stress can be obtained through mode-conversion ultrasonic technology [18]. In summary, the bi-mode wave method could simplify the calibration steps of bolts with different clamping lengths. However, ultrasonic transducers of both longitudinal and shear waves should be used for analysis, which makes the measurement more complicated. The previous single-mode wave method has a simple measurement process, but it is difficult to measure the bolt load under different clamping lengths. Therefore, it is necessary to develop a new clamping length correction method by using a sensor with high measuring accuracy, which not only avoids the complicated calculation process but also realizes the high precision measurement of the axial load of bolts under different clamping lengths.

This paper proposes a novel axial load measurement method by simplifying the calibration using an ultrasonic film sensor. The piezoelectric film sensor can be deposited on the bolt end by metallurgical bonding [19]. Then, the measurement error induced by the couplant can be avoided [20]. Therefore, the single longitudinal wave method has the potential to simplify the experimental process and avoid the error introduced by multiple calculations. Using the physical vapor deposition method, we fabricated high-frequency ZnO film sensors on 40CrNiMo bolts. Based on the ZnO film sensor, the influence of the clamping length on the relationship between axial load and TOF was systematically studied. Then, a simplified calibration method was developed to measure the axial load of the same bolt with different clamping lengths. This method used only single-mode waves and could avoid complicated data processing and calibration steps.

## 2. Clamping Length Correction Method

The propagation velocity of an ultrasonic wave in the bolt varies with changes in the stress inside the bolt [7], and the existence of preload causes tensile stress in the upward direction of the bolt axis. When there is no residual stress and the material is homogeneous and isotropic, the propagation velocity of an ultrasonic wave inside the bolt can be written as:(1)$$v^{\sigma }=v^{0}\left(1+C^{L}\sigma \right),$$
where $v^{0}$ is the ultrasonic velocity of the longitudinal wave without axial stress; $v^{\sigma }$ represents the actual propagation velocity of ultrasonic longitudinal wave under the axial load; and σ is the axial stress of the bolt, which can be written as:(2)$$\sigma =\frac{F}{S_{e}},$$
where F is the axial load; $S_{e}$ is the effective cross-sectional area of the bolt; and $C^{L}$ is the acoustic elastic coefficient of the longitudinal wave, which is related to the bolt material and can be expressed as:(3)$$C^{L}=\frac{\lambda +2l+\left(\mu +\lambda \right)\left(4\lambda +10\mu +4m\right)/\mu }{2\left(\lambda +2\mu \right)\left(3\lambda +2\mu \right)}$$
where λ and μ are the second-order Lame’s constants, and l and m are the third-order Murnaghan’s constants.

For bolts used to fasten connectors, the simplified structure is shown in Figure 1. When applying preload to the nut, only a part of the bolt structure is subjected to axial stress caused by the preloading force. Therefore, it is considered that the total length L of the bolt at this time is composed of the effective length $L_{e}$ and the ineffective part $L_{0}$ without stress, as given by:(4)$$L=L_{e}+L_{0}$$

When no axial load is applied, the time $t_{0}$ of a longitudinal wave propagating inside the bolt can be written as:(5)$$t_{0}=\frac{2L}{v^{0}}$$

Under the pre-tightening load, the part of the area subjected to axial stress will deform, resulting in the total length of the bolt changing. The total bolt length $L_{\sigma }$ can be given by:(6)$$L_{\sigma }=L_{e}\left(1+E^{-1}\sigma \right)+L_{0},$$
where E is the Young’s modulus of the bolt material.

Therefore, the actual propagation time $t_{\sigma }$ of an ultrasonic longitudinal wave inside the bolt under axial load can be written as:(7)$$t_{\sigma }=\frac{2L_{e}\left(1+E^{-1}\sigma \right)}{v^{0}\left(1+C^{L}\sigma \right)}+\frac{2L_{0}}{v^{0}}$$

By contrast with Equation (5), the TOF variation Δt becomes:(8)$$\Delta t=\frac{2L_{e}\left(E^{-1}-C^{L}\right)\sigma }{v^{0}\left(1+C^{L}\sigma \right)\left(1+E^{-1}\sigma \right)}$$

Therefore, the influence of axial load on TOF variation in the bolt can be approximated as the following relation:(9)$$\Delta t\approx \frac{2\left(E^{-1}-C^{L}\right)L_{e}F}{v^{0}S_{e}}$$

When a bolt is pre-tightened, the bolt clamping length $L_{c}$ remains unchanged. In practical scenarios, $L_{c}$ can be measured. For the same bolt:(10)$$L_{e}=L_{c}+\Delta L$$

Under different assembly conditions, the effective clamping length can be obtained by adding ΔL to $L_{c}$. It is generally believed that ΔL consists mainly of two parts, the threaded area engaged with the nut for the transfer of axial load and the part of the bolt head that bears tensile stress. It can be considered that ΔL is approximately constant for the same bolt with different clamping lengths. But in actual measurement, we found that the general calculation equation of ΔL could not be obtained directly from the shape and size of the bolt, and the corresponding ΔL of each bolt should be measured by calibration experiments.

According to the Equations (9) and (10), this can be rewritten as:(11)$$\Delta t=\frac{2\left(L_{c}+\Delta L\right)\left(E^{-1}-C^{L}\right)F}{v^{0}S_{e}}$$

In this equation, $v^{0}S_{e}/2\left(E^{-1}-C^{L}\right)$ depends only on the size and material of the bolt, which can be obtained by calibration experiments. According to this relationship, ΔL at room temperature was obtained by calibration experiments for each bolt and $v^{0}S_{e}/2\left(E^{-1}-C^{L}\right)$ was calculated. The value of axial load could be calculated by measuring the change in TOF and clamping length of bolt assembly.

In the calibration experiment, it is necessary to measure the relationship between the variation in TOF and axial load F under different clamping lengths $L_{c}$, then take the derivative of Equation (11) with respect to F in order to obtain the slope under each clamping length:(12)$$k_{i}=\frac{2(L_{ci}+\Delta L)\left(E^{-1}-C^{L}\right)}{v^{0}S_{e}},$$
where $L_{ci}$ is the configuration of different clamping lengths, and $k_{i}$ is the slope of the relationship between TOF variation and axial load under different clamping lengths. Since $v^{0}S_{e}/2\left(E^{-1}-C^{L}\right)$ and ΔL of the same bolt are fixed, they can be calculated by repeated tensile tests under different clamping lengths.

For the same bolt:(13)$$\frac{L_{ci}+\Delta L}{k_{i}}=\frac{L_{cj}+\Delta L}{k_{j}},$$

i and j are configured for different clamping lengths. The ΔL of the bolt can be calculated, and the reliability of the axial stress measurement method can be verified by the calculation results.

## 3. Experiment

### 3.1. Piezoelectric Film Sensor

The film sensor can generate and receive high-frequency ultrasonic longitudinal waves at the bolt end and is not affected by couplant, adhesives, or other factors. During the whole working cycle of the bolt, the echo TOF is only affected by temperature and load changes. As a ceramic sensor with strong stability, it can be used for long-term monitoring of bolt stress changes.

Using the physical vapor deposition method, we fabricated a high-frequency piezoelectric sensor for load measurement using film deposition equipment. The sensor preparation device was mainly composed of a vacuum chamber, power supply, heater, target, substrate, and other parts, using the purity of 99.99% ZnO ceramic target. The ZnO piezoelectric material was a semiconductor with a wide bandgap of about 3.33 eV. It had a high crystalline structure, high piezoelectric coefficient, high electromechanical coupling coefficients, low cost, good charge carrying ability, was easy to deposit on the bolt surface, and could generate ultrasonic longitudinal waves under electric pulse excitation [21,22,23]. The distance between the target and the bolt head was 110 mm. Before placing the bolt into the vacuum chamber, the end of the bolt was cleaned and polished, and the part outside the head was wrapped with tin foil. Then, ion etching was carried out in an Ar atmosphere to clean the bolt surface and improve the bonding force between the sensor and the bolt, thus improving sensor stability and prolonging its service life.

By changing the sputtering power, working pressure, and target base distance, the growth behavior of the piezoelectric coating could be changed. We found that the piezoelectric coating had the best piezoelectric effect when the sputtering power was 350 W, the sputtering pressure was 2.5 Pa, and the target substrate distance was 140 mm. At this point, the coating had a high c-axis orientation with only a (002) diffraction peak and it was easy to generate ultrasonic longitudinal wave. At the same time, in order to reduce shear wave generation and enhance the longitudinal wave signal as much as possible, the bolt was placed in the center and middle position. On this basis, the RF sputtering method and arc discharge method were used to prepare piezoelectric film sensor bolts. The bolt sensor diagram is shown in Figure 2a. The protective layer was a high-entropy alloy material with high resistance, and the electrode layer was Ti or Ag. Under the excitation of electric pulse, the piezoelectric film region covered by the electrode could excite ultrasonic longitudinal waves.

The cross-sectional morphology of the samples was examined using a MIRA3 scanning electron microscope (SEM) operated at 20 kV (MIRA3LMH, TESCAN, Brno, Czech Republic). Figure 2b shows the cross-section of the sensor and the bolt end. The sputtered ZnO layer was 17.4 μm thick, above which was the SiO_2_ protective layer and Ti electrode layer, and the total thickness of the sensor was 23.6 μm. By adjusting the deposition time and controlling the thickness of the ZnO piezoelectric layer, the sensor structure with high-frequency ultrasonic longitudinal wave excitation ability was obtained.

We prepared piezoelectric film sensors for two kinds of wind turbine tower bolts. The bottom of the wind turbine tower bolt faces the engine room during operation and can be used to prepare sensors and receive measurement data. The bolt was made of 40CrNiMo alloy with the following composition: Ni 1.5%, Cr 0.7%, Mo 0.2%, and C 0.4%. Table 1 shows the specifications of the two kinds of bolts and their sensors.

After the preparation of the protective layer and electrode, a thin film structure with ultrasonic sensor function was obtained. Ultrasonic waves were generated by ZnO piezoelectric films excited by the ultrasonic measuring instrument. The ultrasonic measuring instrument mainly comprised a data acquisition card (PCI-5114, NI, Austin, TX, USA), pulser/receiver (DPR300, Imaginant, New York City, NY, USA), and control computer. The maximum sample frequency of the data acquisition card was 250 MHz.

The top electrode layer of the sensor was connected to the cable of the pulser/receiver by a magnetic suction contact probe. The external electrode layer was deposited directly on the bolt matrix and connected to the housing of the pulser/receiver, thus forming a loop between the pulser/receiver and the sensor. The pulser/receiver provided the functions of generating, amplifying, filtering, and receiving the electric pulse signal. When the pulser/receiver contacted the electrode, an electric field was formed at both ends of the piezoelectric coating. The piezoelectric sensor converted the electric pulse signal into an ultrasonic signal, and after the ultrasonic signal was reflected on the bottom of the bolt, the sensor converted the ultrasonic echo signal into an electrical signal. The pulser/receiver received each ultrasonic echo and transmitted it to the processing PC for subsequent processing. The software part was compiled by LabVIEW, including the parameter setting module, the temperature and ultrasonic signal acquisition and storage module, and the preload calculation module.

### 3.2. Ultrasonic Test Experiment

The aim of this experiment was to test the ultrasonic performance of a piezoelectric film sensor and verify the method of bolt axial load measurement based on the ultrasonic longitudinal wave method. In order to accurately control the bolt preload, as shown in Figure 3, tensile tests were carried out on annealed bolts with a stretcher (AGX-V, SHIMADZU, Kyoto, Japan) under different clamping lengths. Tensile load was used to replace the bolt axial preload, and contact cables were used to receive piezoelectric wafer signals at the bolt head. Five commonly used clamping length ranges were selected for two kinds of wind turbine tower bolts, and tensile calibration was carried out under each clamping length. The axial load range was 0–300 kN and the loading gradient was 75 kN. The tensile stress corresponding to this application load was less than 50% of the yield strength of the bolt, thus ensuring that no plastic strain occurred during the experiment, and the bolt loading and unloading speed was 500 N/s. In the reciprocating loading test, the signals were collected separately and averaged. Standard washers fitted to bolts were used to ensure consistency between the calibration experiments and actual working conditions. The axial loading error of the stretcher could be determined to be less than 0.1% by reciprocating loading and unloading. The nut was screwed into the corresponding position of each clamping length in advance, the clamping length was measured after the bolt was subjected to axial load, then the changes in the bolt axial load and TOF were recorded. The room temperature was maintained at 16 °C during the experiment.

During the experiment, the gain of the pulser/receiver was set to 40 dB, the excitation voltage amplitude was 200 V, the excitation signal was a Gaussian function, the pulse width was 70 ns, and the pulse repetition frequency was 5 kHz. The signal with a frequency between 2.5 and 50 MHz was received through the filter, and the sampling frequency was 250 MHz. An appropriate average number of signals was set to reduce the acquisition error, and the TOF fluctuation w less than 0.2 ns under the same load.

## 4. Results and Discussion

### 4.1. Characterization Results of Piezoelectric Film Sensor

An ultrasonic measuring instrument was used to characterize the ultrasonic performance of the sensor. Figure 4a shows an M42 wind turbine tower bolt with the piezoelectric film sensor. Multiple echo signals could be received through the ultrasonic acquisition instrument. Figure 4b shows the ultrasonic signal and echo excited by the piezoelectric sensor at the top of the bolt. According to the bolt length and the velocity of the longitudinal wave in the 40CrNiMo material, it could be determined that the first echo of the longitudinal wave was about 91.316 μs, which is marked in the figure, and the subsequent longitudinal wave echoes were separated by a fixed period. Since the echo signal was affected by the propagation time of the electromagnetic wave in the cable, the difference between the two longitudinal wave echoes was used as the round-trip time in the bolt. Ultrasonic shear waves could also be observed between the two echoes, which was caused by the deviation of the bolt from the central position or an inappropriate Ar/O_2_ ratio during the deposition process.

The analysis software could directly identify the pulse echo and the maximum value of each echo signal during each measurement, and it calculated the time for the longitudinal wave to travel back and forth through the bolt. Cubic spline interpolation was used to improve the resolution of data acquisition. After the axial load was applied by the stretcher, the variation in the longitudinal wave TOF was obtained by cross-correlation analysis with the stored initial waveform. At the same time, the collected longitudinal wave was analyzed, and the primary longitudinal wave with its frequency spectrum of sensors was obtained by fast Fourier transform (FFT). A relatively representative set of bolts are shown in Figure 5a,b. The pulse–echo response of the thin film sensor was concentrated at 25 MHz with a −6 dB bandwidth of 47% for the M42 bolt and 33 MHz with a −6 dB bandwidth of 69% for the M48 bolt.

### 4.2. Influence of Clamping Length on Axial Load Measurement

Through the tensile measurement experiments, the relationship between Δt and the axial load F under different clamping lengths $L_{c}$ was measured, as shown in Figure 6. The specific clamping lengths of the two bolts are shown in the figure. The TOF had an obvious linear relationship with the change in axial load, and it can be seen from the figure that the slope k varied greatly under different clamping lengths. Therefore, different clamping lengths could induce huge errors in bolt load calibration.

The slope k under each clamping length was obtained by linear regression, then the corresponding relationship between different clamping lengths and slope k was obtained, as shown in Figure 7, and linear fitting was carried out. The regression equation of the M42 bolt could be expressed as $k=0.00467\times L_{c}+0.03484$, where the unit of $L_{c}$ is mm. The coefficient of determination $R^{2}$ was 0.999. Additionally, the regression equation of the M48 bolt could be expressed as $k=0.00319\times L_{c}+0.12820$, with a coefficient of determination $R^{2}$ of 0.999.

The slope k under different clamping lengths showed good linearity with $L_{c}$, which corresponded to the derivation process from Equation (11) to Equation (12), and verified the conclusion of Equation (12), that is, slope k is linearly correlated with $L_{c}$. According to Equation (13), ΔL could be calculated for every two sets of calibrations. The average value of ΔL, stored as the calibration parameters of this bolt, and the corresponding $v^{0}S_{e}/2\left(E^{-1}-C^{L}\right)$ for each clamping length configuration were obtained by Equation (12). The calculation results of ΔL and $v^{0}S_{e}/2\left(E^{-1}-C^{L}\right)$ are shown in Table 2.

Almost the same calculation results of ΔL and $v^{0}S_{e}/2\left(E^{-1}-C^{L}\right)$ were obtained by tensile tests under different clamping lengths, which verified our measurement method and process. The average value was taken as the calibration parameter in the bolt measurement process. By measuring the effective cross-sectional area $S_{e}$ of the bolt and calculating the initial ultrasonic velocity $v^{0}$ in the bolt, the $E^{-1}-C^{L}$ value of the bolt of the same material could be obtained, but it was of no practical significance for the bolt load measurement. In the axial stress measurement, the value could be regarded as a fixed value, so there was no need to measure $S_{e}$ of the bolt and $v^{0}$ when calibrated bolts had different clamping lengths and axial loads. The ultrasonic longitudinal wave TOF could be obtained by the ultrasonic acquisition instrument, and the bolt clamping length could be measured. The axial load could be determined by Equation (11) through the calibration parameters stored in the computer.

For the two kinds of wind turbine tower bolt, the errors of preloading force measurement under different clamping lengths are shown in the Figure 8. The maximum relative errors were 2.7% for the M42 bolt and 1.3% for the M48 bolt. The main error in the measurement of axial load comes from the difference between effective stress length and the calculated effective clamping length. Through the calibration experiments under different clamping length configurations, a consistent ΔL was obtained, which ensured the high precision of load measurement. At the same time, high-precision measurement of TOF was realized by the film sensor, and the error of the slope calculation in the load calibration was reduced. Accurate length measurement is the premise of load measurement accuracy; load measurement error will increase for smaller bolts.

The effective cross-sectional area of the thread area is different from that of the area without thread. When the clamping length changes, the average effective cross-sectional area of the area under stress also changes, but the effective cross-sectional area is regarded as a fixed value in the calculation process, which indirectly lead to difficulties in establishing the relationship between ΔL and geometrical dimensions such as the bolt diameter.

Finally, the applicability of this method to calculate $L_{e}$ was studied. For the M42 bolts used in this experiment, the calculation results of ΔL would not lead to significant errors in load measurement when the clamping length was 165–210 mm. When the nut was screwed into the top or bottom area of the thread, the error in load measurement would increase significantly. This was due to the fact that the stress state of the nuts at both ends of the thread was different from that at other positions, which should be avoided in engineering applications.

## 5. Conclusions

In this paper, an effective clamping length correction method based on a piezoelectric film sensor was proposed to measure the axial load of bolts with different clamping lengths. The relationship between longitudinal wave TOF and axial load under different clamping lengths was established, thus correcting the difference between the effective clamping length and the actual clamping length and effectively simplifying the calibration experiment process. High-frequency ZnO piezoelectric film sensors were fabricated to improve the accuracy of TOF measurements and eliminate the influence of the couplant. Two bolts with piezoelectric film sensors were used to verify the practicability of this measurement method. The center frequency of the film sensor was 25 MHz, and the measurement error of the axial load of the two bolts was less than 2.7%. The experimental results showed that the proposed axial load measurement method could achieve high-precision measurement of axial load under different clamping lengths.

The present work focused on measuring the axial load of bolts. However, bolts can be subjected to multidirectional loads during their service life. Simultaneous measurement of both axial and transversal load is a challenge that will be addressed in future work. The further research plan is to develop an array of thin-film sensors to measure unevenly distributed loads on bolts.

## Figures and Tables

**Figure 1 sensors-23-03712-f001:**
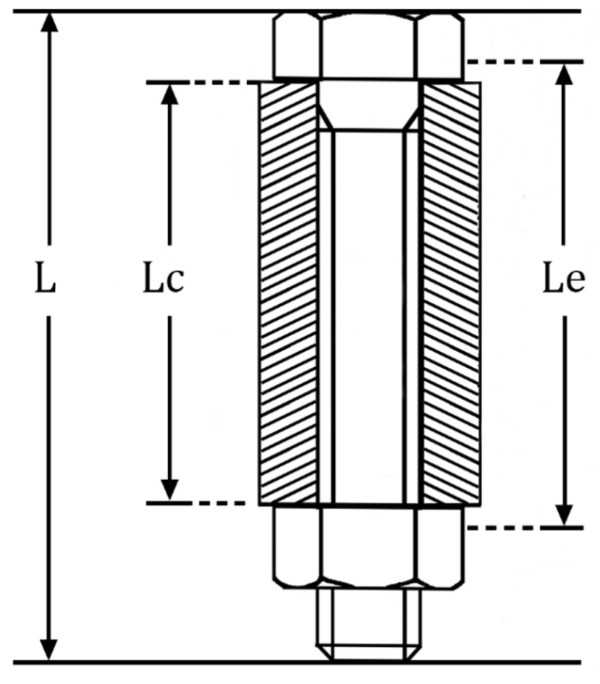
Bolt joint structure.

**Figure 2 sensors-23-03712-f002:**
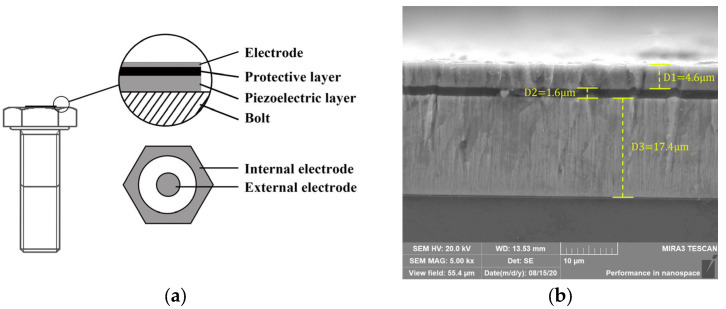
(**a**) Sensor structure diagram. (**b**) Film sensor cross-section.

**Figure 3 sensors-23-03712-f003:**
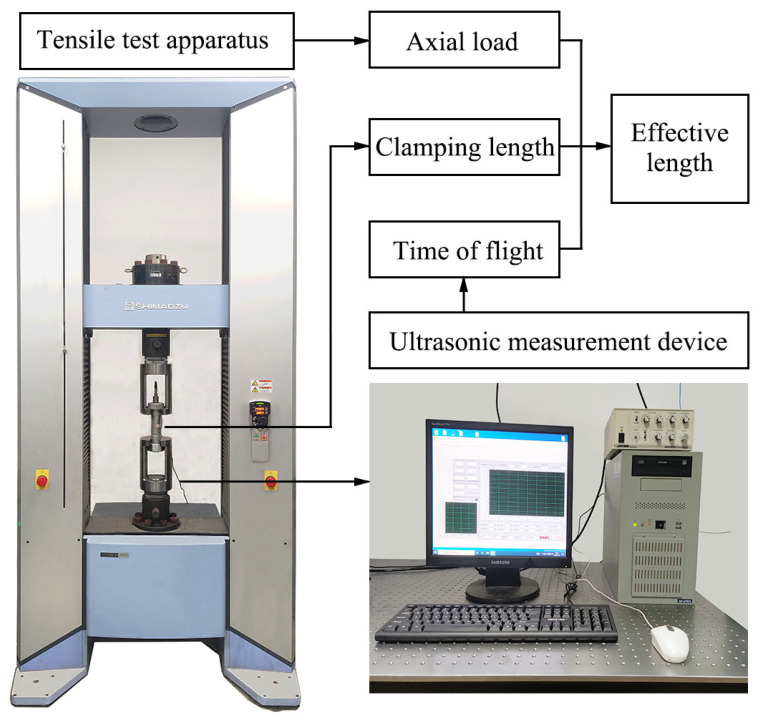
Bolt axial tension measurement experiment process.

**Figure 4 sensors-23-03712-f004:**
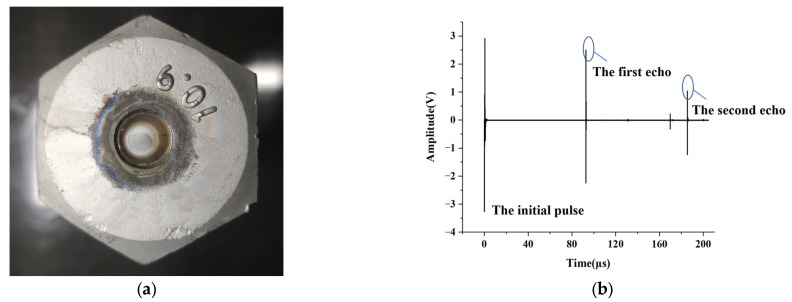
(**a**) M42 wind turbine tower bolt with piezoelectric film sensor. (**b**) Film sensor echo signal.

**Figure 5 sensors-23-03712-f005:**
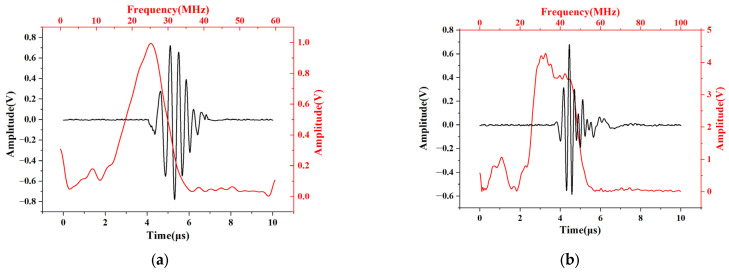
Pulse–echo waveform and frequency spectrum for (**a**) M42 bolt; (**b**) M48 bolt.

**Figure 6 sensors-23-03712-f006:**
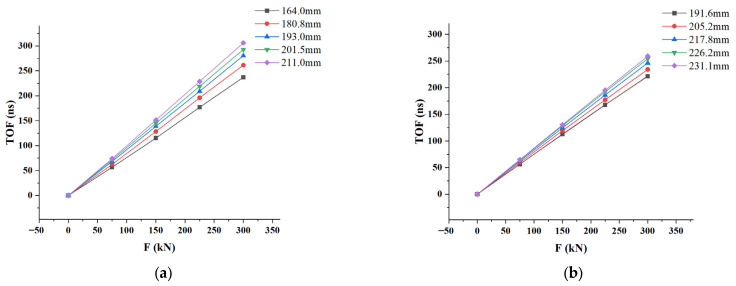
Relationship between Δt and the axial load under different clamping lengths for (**a**) M42 bolt; (**b**) M48 bolt.

**Figure 7 sensors-23-03712-f007:**
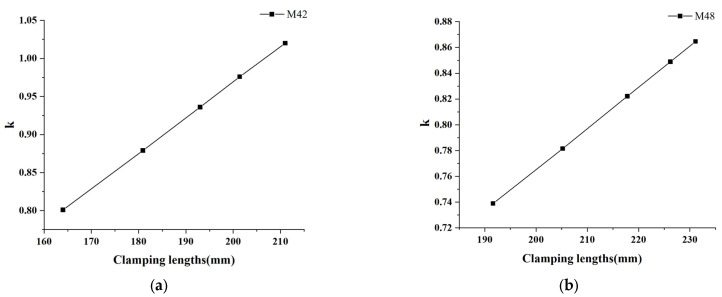
k at different clamping lengths for (**a**) M42 bolt; (**b**) M48 bolt.

**Figure 8 sensors-23-03712-f008:**
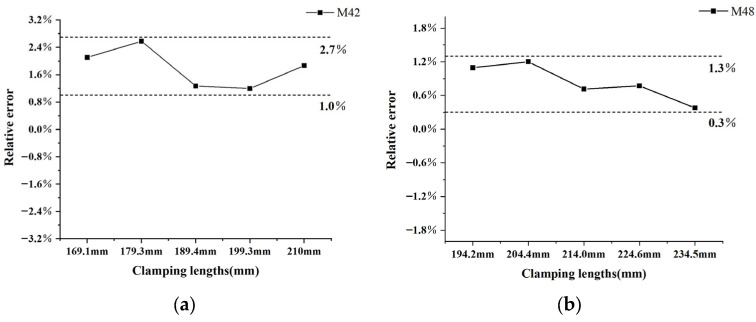
Measurement error of preload under different clamping lengths for (**a**) M42 bolt; (**b**) M48 bolt.

**Table 1 sensors-23-03712-t001:** Specification for bolts with film sensor.

Bolt Type	Total Length/(mm)	Thread Area/(mm)	Material	Sensor Size
M42	268	160–268	40CrNiMo	ϕ16
M48	303	182–303	40CrNiMo	ϕ16

**Table 2 sensors-23-03712-t002:** ΔL  and $v^{0}S_{e}/2\left(E^{-1}-C^{L}\right)$ in different clamping length configurations.

M42	164 mm	180.8 mm	193.0 mm	201.5 mm	211.0 mm	Average
$\Delta L\;\left(\mathrm{mm}\right)$	—	7.86	7.84	6.13	7.41	7.31
$\frac{v^{0}S_{e}}{2\left(E^{-1}-C^{L}\right)}\times 10^{-11}\;\left(N\cdot m/s\right)$	4.671	4.673	4.673	4.675	4.673	4.673
M48	191.6 mm	205.2 mm	217.8 mm	226.2 mm	231.1 mm	Average
$\Delta L\;\left(\mathrm{mm}\right)$	—	44.43	37.21	40.31	38.18	40.03
$\frac{v^{0}S_{e}}{2\left(E^{-1}-C^{L}\right)}\times 10^{-11}\;\left(N\cdot m/s\right)$	3.191	3.189	3.188	3.189	3.191	3.190

## Data Availability

Not applicable.

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
