# Peer review of "Axial Load Measurement of Bolts with Different Clamping Lengths Based on High-Frequency Ultrasonic ZnO Film Sensor"

_sensors, 2023, doi:10.3390/s23073712_

Round 1
Reviewer 1 Report
Title: Axial load measurement of bolt with different clamping lengths based on high frequency ultrasonic ZnO film sensor
This is a well-organized study, however, there are a few things that need to be revised:
Abstract:
Please describe the main results and also describe important applications.
Section 2 should merge into section 3 as section 3.1.
Page 6- The MIRA3 field emission scanning electron microscope (SEM): provide analyzed device information, the same issues should be addressed throughout Section 3.
Correct spelling and sentences: Ti electrode layer, SiO2 protective layer, ZnO piezoelectric layer and bolt matrix can be observed from top to bottom
Check the information: “The bolt is made of 40CrNiMo alloy with composition: Ni 1.5%, Cr 0.7%, Mn 0.6%, and C 0.4%.”
Page 7-
Correct spelling and sentences: “The parameter setting module includes setting sampling depth, frequency division, Min. sample length, average times, etc., which can realize real-time adjustment of”
About - Five commonly used clamping length ranges were selected for two kinds of wind turbine tower bolts, and tensile calibration was carried out under each clamping length: Did the authors repeat the stretch test? A round-robin test is recommended for verification.
Section 4.1- please verified and provide the “Multiple echo singles”: Multiple echo signals can be received through ultrasonic acquisition instruments.
Figure 5- Please verify the red line of the FFT curve. The curve, thin film sensor centered at 25 MHz for M42 and at 33MHz for M48 bolt, has repeated characterization or not? Would any deviation affect the results?
Reviewer 2 Report
In this paper authors suggest employing a piezoelectric film as an ultrasonic transducer for axial load measurements in the calibration process of bolts.
The topic fits well in the scope of Sensors journal, the paper is well written and it might be of interest to the readership. However, there is some place for improvements in the domain of the scientific soundness.
In general, I suggest adding some text with a greater emphasis on what is in this paper scientifically new and improving the clarity of explanations already given.
In introduction, elaborate more on problems with current solutions for axial load measurement and on the scientific advance behind your solution for the improvements in the axial load measurement method.
Further, elaborate more the transition from equation (8) to equation (9). Why is the approximation appropriate, what are typical numerical values for neglected terms?
Then, in the first sentence of the last paragraph in section 3.1: elaborate more on that connection and the type of bond.
Additionally, Figure 4 would be easier to read if it would have been given with bigger font (the same size as the font in the figure caption) and sharper lines. The same problem poses Figure 8, too small labels and low readability.
Some more clarifications are suggested for the first sentence in the paragraph below figure 7. It should be rephrased or given with better arguments. Equation (12) is not of statistical nature, so it does not show correlation.
My last suggestion is on references. The advice is to add some more of newer, recent references so that the topic appears to be in the spotlight of the current research.
Thank you for contributing with your work,
With kind regards...
Reviewer 3 Report
This paper presents a delicate design upon ultrasonic detection based on ZnO thin films. The proposed model is proven to be effective for decreasing the measurement error below 2.7% in this experiment. The work is sound and interesting. Before its application, some necessary revision shall be done:
[1] The introduction part, paragraph 5 and 6 shall be merged, instead of both starting as "this paper". It reads redundant and unnatural.
[2] The use of ZnO film can be further elaborated, why this material?
[3] The deposition of ZnO generally includes evaporation and solution deposition, what is author's opinion about these two methods, which one is better in this scenario?
[4] Typos, grammatical mistakes and language fluency should be improved.
